# Performance benchmarking of LLMs on Chinese national medical licensing education: Cross-lingual and question-type effects

Yuxia Tang[1,2]*, Jian Chen[2], Shouju Wang[1]*

1 Department of Radiology, the First Affiliated Hospital with Nanjing Medical University, Nanjing, China,
2 Department of Radiology, Chongqing Hospital of Jiangsu Province Hospital, Chongqing, China

☉ These authors contributed equally to this work.
* tangyuxia@njmu.edu.cn (YT); shouju.wang@gmail.com (SW)

## Abstract

### Background

The cross-lingual and question-type variations affecting large language models (LLMs) accuracy on the Chinese national medical licensing educations remain insufficiently explored.

### Methods

In this cross-sectional study (May 13–20, 2025), 396 educational questions (198 English–Chinese pairs) were extracted from the Chinese national medical licensing examination. ChatGPT-4o, ChatGPT-o3, Gemini-2.5-pro, Deepseek-V3, Deepseek-R1, and Doubao-1.5-pro were prompted to provide answers. Responses were compared against reference answers, and accuracy was computed for three question types: basic knowledge (Type A), case analysis (Type B), and integrative judgment (Type C).

### Results

Across all question types and languages, Doubao-1.5-pro achieved the highest accuracy at $92.0\% \pm 1.3\%$, whereas ChatGPT-4o had the lowest accuracy at $82.8\% \pm 3.7\%$. There was a significant main effect of question type ($P = 0.0038$) but no main effect of language ($P = 0.56$). Post hoc tests confirmed that Type A performance exceeded Types B and C ($P < 0.01$), while B vs. C did not differ. Among the models, Doubao-1.5-pro, Deepseek-R1, and Deepseek-V3 demonstrated notable cross-lingual stability, with accuracy differences between Chinese and English versions remaining below 5%.

### Conclusion

The question type was a key factor affecting LLMs performance on Chinese medical licensing exam questions, whereas language had no significant impact.

**Data availability statement:** All relevant data are within the manuscript files.

**Funding:** This work was supported by Chongqing Natural Science Foundation (CSTB2024NSCQ-KJFZMSX0065) and the Foundation of Nanjing Medical University (No. JX1231600602), Jiangsu Province Hospital (TS202401, CZ0121002010039). There was no additional external funding received for this study.

**Competing interests:** The authors have declared that no competing interests exist.

Doubao-1.5-pro, Deepseek-R1, and Deepseek-V3 demonstrated particularly strong cross-lingual consistency. These findings point to the potential value of specialized LLMs for enhancing medical education in China.

## Introduction

Artificial intelligence has advanced rapidly in recent years, leading to the integration of generative large language models (LLMs) into essential healthcare services. These models are now used in applications from clinical knowledge management to telemedicine consultations. The medical field, with its complex knowledge requirements and critical decision-making contexts, provides an important opportunity to evaluate how well artificial intelligence can perform in real-world scenarios. The study in medical education demonstrates that GPT-4 can improve medical students' diagnostic accuracy by 19% [1]. Furthermore, extensive research shows that LLMs have significant clinical knowledge, performing at expert levels across various medical specialties [2–4].

The Chinese medical licensing examination is one of the world's largest clinical competency assessments, using various question types to evaluate foundational knowledge, clinical reasoning, and practical application skills. While researchers have investigated large language model performance on the United States medical licensing exams and similar assessments [5,6], there is a lack of comprehensive research examining how LLMs perform when evaluated across different languages and question types [7].

This research presents a new mixed-format evaluation framework for Chinese medical licensing examination, utilizing six major large language models: OpenAI's ChatGPT-o3 and ChatGPT-4o, Google's Gemini-2.5-pro, DeepSeek's R1 and V3 models, and ByteDance's Doubao-1.5-pro [8,9]. The purpose of this study is to investigate the performance of LLMs on Chinese medical licensing examination, with the goal of providing valuable insights for medical education and training.

## Materials and methods

### Questions extraction

The Chinese medical licensing examination utilizes three standardized formats: Type A (independent items), Type B (shared-stem items), and Type C (shared-option items). While Type A items focus on foundational knowledge retrieval, Type B and Type C formats introduce significant contextual complexity. Type B items require the model to sustain clinical reasoning across a multi-stage vignette, while Type C items assess its precision in differential diagnosis when faced with a static set of competing distractors. Since official examination papers are not publicly released, all items for this study were sourced from widely recognized, publicly available preparatory materials provided by Beijing Medical Examination Assistance Technology Co., Ltd. We selected materials from 2020−2022 to ensure the inclusion of the most comprehensive and contemporary collections with verified answer keys. To ensure the statistical

validity of this sample size, we calculated that 198 items drawn from an annual population of approximately 600 questions provide a margin of error of approximately ±5.8% at a 95% confidence interval (assuming a 50% response accuracy for maximum variance). This degree of precision is widely considered sufficient for the evaluative analysis of LLMs. To maintain consistency for cross-lingual evaluation, only text-based items were included, while questions containing images, tables, or chemical formulas were excluded. From this pool of available text-based items, we employed a stratified random sampling approach to select 198 questions, strictly preserving the original distribution of Type A, B, and C formats. This resulted in a dataset comprising 76 Type A, 62 Type B, and 60 Type C items. The English versions were generated through a standardized forward-translation process conducted by two authors (YX Tang and SJ Wang), both of whom possess over ten years of clinical experience and have completed more than one year of formal medical training in the United States. The study was conducted from May 13–20, 2025. The LLMs include ChatGPT-o3 (released April 17, 2025), ChatGPT-4o (released May 13, 2025), Gemini-2.5-pro (released March 25, 2025), Doubao-1.5-pro (released January 22, 2025), Deepseek R1 (released January 20, 2025), and Deepseek V3 (released December 26, 2024).

## Response generation

All models were accessed through their official web interfaces using default parameters. Below are the input methods for different question types: for type A questions, each question was entered in a fresh conversation session; for type B questions, questions sharing the same stem were entered together in a single session, followed by a new conversation session for the next set of questions with a different shared stem. For type C questions, questions sharing the same set of options were entered together in one session, with a new conversation session initiated for the next group of questions with different shared options. During the experiment, research assistants used a standardized prompt: "You should act as an educational expert of Medicine. When faced with a choice question of type A, where each question has only one best answer, you need to provide the optimal response. The question is "The primary damage to DNA by ultraviolet rays is the formation of: A. Base deletion; B. Base insertion; C. Base substitution; D. Thymine dimers; E. Phosphodiester bond cleavage". You are required to present your answer in the following structure: Response: Answer." (Fig 1). Two medically trained assistants independently verified each model's answer against the official key, labeling responses as correct or incorrect.

## Statistical analysis

Statistical analyses began with descriptive summaries of each model's mean accuracy across every combination of question type (A, B, C) and language (Chinese, English). To formally assess differences, a two-way repeated-measures ANOVA was conducted with mean accuracy as the dependent variable, models as the subject factor, and question type (three levels) and language (two levels) as within-subject factors; this allowed testing of the main effects of question type

**Fig 1. Schematic diagram of the English version of the structured prompt template.** Yellow-colored placeholder denotes the question type (A, B, C), while blue-colored placeholder denotes the question stem. Type A (Single-Best-Answer Questions): Direct, standalone multiple-choice items that test the retrieval of basic medical facts. Type B (Case-Based Analysis Questions): Full-case prompts—including chief complaint, medical history, and diagnostic findings—designed to evaluate a candidate's clinical reasoning processes. Type C (Shared-Option Questions): Multi-part items that present a common set of five diagnoses and then pose sequential questions on pathogen identification and radiologic feature matching, thereby probing candidates' ability to transfer and apply their knowledge in multidimensional diagnostic scenarios.

and language as well as their interaction. Following a significant main effect of question type, post hoc paired t-tests were performed between each pair of question types (A vs. B, A vs. C, B vs. C), and a separate paired t-test compared Chinese versus English, with all resulting p-values adjusted via the Bonferroni correction. All statistical analyses were performed in R (version 4.2) and figures were produced with ggplot2.

## Results

Each model's overall accuracy (mean ± SD) was calculated across all question types and languages: ChatGPT-4o 82.8% ± 3.7%, ChatGPT-o3 86.1% ± 2.5%, Deepseek-V3 89.8% ± 1.4%, Deepseek-R1 91.0% ± 1.5%, Doubao-1.5-pro 92.0% ± 1.3%, and Gemini-2.5-pro 88.7% ± 1.7% (Fig 2). All models performed most reliably on Type A questions, maintaining accuracy above 88% on average. Type C questions, however, showed the greatest variability in performance, ranging from 72% to 95%. ChatGPT-4o scored 88.2% on Type A, 82.3% on Type B, and 71.7% on Type C in Chinese, while in English the scores were 97.4%, 77.4%, and 80.0% respectively. Deepseek-R1 stood out with particularly strong performance on Type C questions, achieving 95.0% accuracy in Chinese and 88.3% in English, which was superior to most other models in our evaluation. Notably, Doubao-1.5-pro, Deepseek-R1, and Deepseek-V3 all showed strong cross-lingual stability, with language-induced accuracy differences under 5%. For Type C questions, the performance difference between Deepseek-R1 (the top model) and ChatGPT-4o (the lowest-performing model) was quite large—almost 25% (Fig 3).

There was a significant main effect of question type ($F_{(2,10)} = 10.22$, $P = 0.0038$), but no significant effect of language ($F_{(1,5)} = 0.39$, $P = 0.56$) (Fig 4). A marginal interaction between question type and language was observed ($F_{(2,10)} = 3.57$, $P = 0.068$). The performance of Type A was significantly better than both Type B ($P = 0.002$) and Type C ($P = 0.025$).

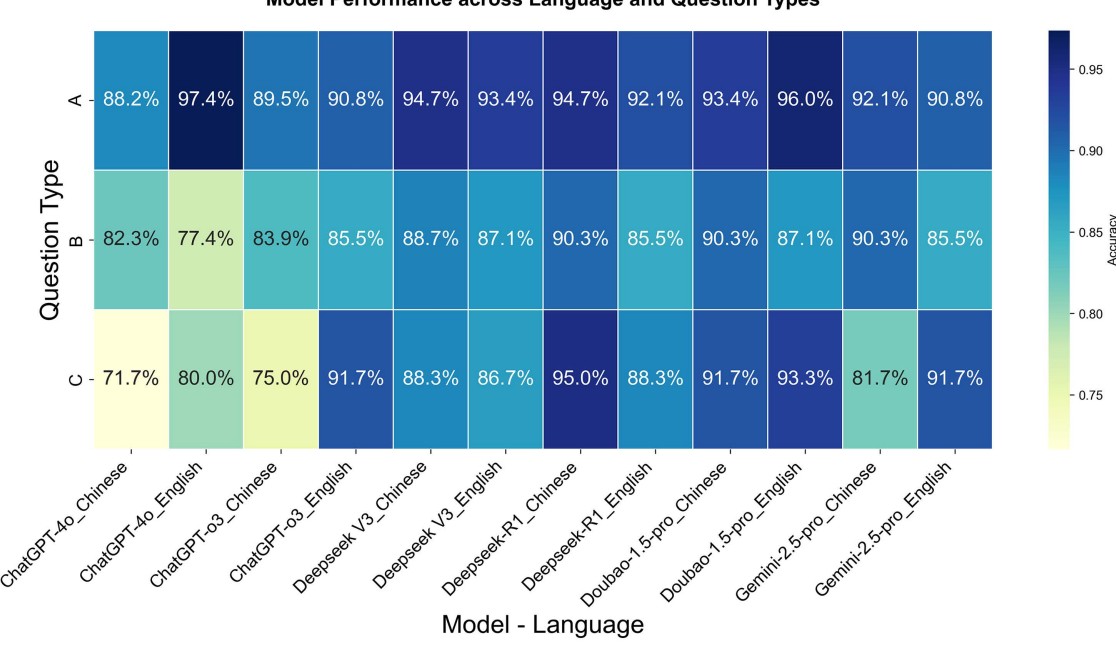

**Fig 2. Performance evaluation of generative language models on Chinese medical licensing examinations.Heatmap displays accuracy (%) of six large language models across three bilingual question formats: Type A, Type B and Type C. Columns represent models, while rows denote question-type-language combinations.** Color gradient reflects model competency levels, with deep blue (>90%) indicating clinical reliability thresholds and yellow (<75%) signaling high-risk zones requiring human oversight.

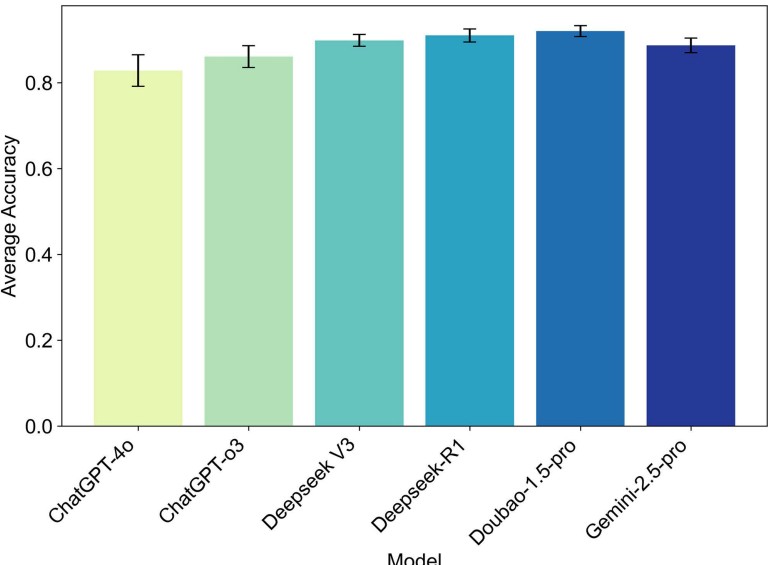

**Fig 3. The average accuracy, along with its standard error range, of different large language models (Large language models) in answering all 198 questions.**

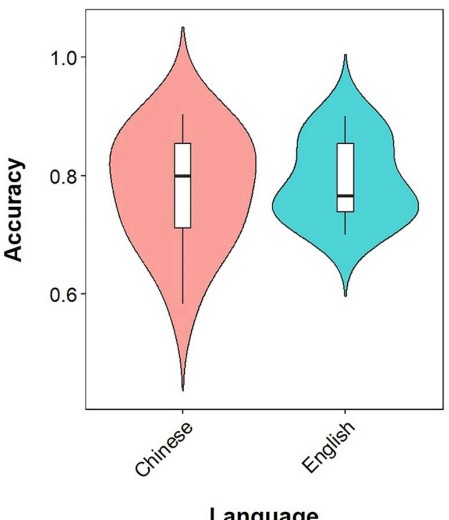

**Fig 4. A violin plot comparing accuracy distributions between Chinese and English in educational question answering.** The central line indicates the median score, with box edges marking the 25th and 75th percentiles, defining the interquartile range (IQR) of the distributions.

However, neither the Type B vs. Type C comparison nor the Chinese vs. English comparison reached statistical significance after correction.

## Discussion

This study systematically evaluated the performance of six LLMs on Chinese medical licensing examination questions across both Chinese and English versions and different question types. Our comprehensive analysis yielded several

key findings. Question type emerged as the primary factor influencing model performance, with a significant main effect observed. All models performed most consistently on basic knowledge questions and complex integrative questions revealed substantial performance variability. Language had minimal impact on overall performance, with no significant main effect detected. This stability suggests these models have developed robust semantic representations that transcend surface linguistic differences.

We also found that complex questions have the ability to distinguish between different models, which is in line with cognitive theories. Cognitive theories suggest that higher – order thinking tasks can better highlight differences in reasoning abilities. Several studies have shown that models with specialized reasoning capabilities, such as those using chain – of – thought prompting or other reasoning – enhancement techniques, usually perform better than general – purpose models when dealing with complex medical questions [10,11]. Our research also made a similar discovery. Models like Doubao-1.5-pro and Deepseek-R1, which may adopt more advanced reasoning architectures, performed particularly well on complex questions [12,13]. Our research further expands on previous research findings. We found that as the complexity of questions increases, the performance gap between different models becomes larger. For example, on Type C questions, the difference in accuracy between Deepseek-R1 and ChatGPT-4o was nearly 25%. This indicates that complex integrative tasks may be very useful for evaluating and comparing the reasoning abilities of different LLMs in the medical field.

Our findings align with and extend previous research evaluating LLM performance on medical licensing examinations in China and internationally. Prior studies have shown that LLMs generally perform well on foundational recall questions but exhibit greater variability on complex reasoning tasks, consistent with our observation of significant question-type effect [12,14]. Other evaluations of Chinese medical licensing examinations similarly reported that English prompts may improve performance on basic knowledge tasks but offer limited benefit for higher-order reasoning [15]. Additionally, recent work in dental and specialty examinations demonstrated that cross-lingual performance varies substantially across models, highlighting the importance of bilingual semantic alignment [16]. Our study contributes to this literature by providing a direct cross-lingual comparison across three question types and identifying models with particularly strong cross-lingual stability.

Some previous studies have found that using English prompts can make models do better in basic knowledge tasks [17–19]. But our research results show that this advantage fades when it comes to more complicated clinical reasoning tasks. This is in line with other research which points out that although large English language databases can help with simple questions, they don't offer much help for complex medical reasoning [20,21]. It's really worth noticing that models like Doubao-1.5-pro, Deepseek-R1, and Deepseek-V3 show strong stability across different languages. This finding fits well with other studies that prove models trained with bilingual biomedical data and semantic alignment methods can have great cross-lingual performance [22–24].

This study has several limitations. Although the passing standard for this examination is 60% accuracy, the results of this study do not indicate that these LLMs can pass the real examination. This is because the analysis was based on only 198 questions rather than the complete examination set, and it was limited to text-based single-choice questions, excluding items containing images, tables, chemical formulas, and other non-text modalities that are typically included in the full examination. This study only focused on accuracy, neglecting consistency, rationale, educational value, and clinical validity. Future studies are needed to address these issues. Extending assessments to various dimensions and real-world scenarios will better establish LLMs' utility and reliability in clinical education.

In conclusion, this study provides a comprehensive evaluation of six LLMs on the Chinese national medical licensing examination across languages and question types. A central finding is that question type, rather than language, is the primary determinant of model performance. All models showed consistent accuracy on basic knowledge questions but exhibited substantial variability on complex questions. Among the models evaluated, Doubao-1.5-pro and Deepseek-R1 demonstrated superior overall performance and maintained strong cross-lingual stability, with low language-related accuracy differences. These findings have important implications for the development and deployment of LLMs in medical education.

## Acknowledgments

We would like to express our sincere gratitude to Dr. Chuanbing Wang for his kind assistance in using ChatGPT and Gemini during his stay in the United States in May 2025.

## Author contributions

**Conceptualization:** Yuxia Tang, Jian Chen, Shouju Wang.

**Data curation:** Yuxia Tang, Jian Chen.

**Formal analysis:** Yuxia Tang, Jian Chen.

**Funding acquisition:** Yuxia Tang, Shouju Wang.

**Investigation:** Shouju Wang.

**Methodology:** Yuxia Tang, Jian Chen.

**Project administration:** Yuxia Tang.

**Supervision:** Shouju Wang.

**Validation:** Shouju Wang.

**Writing – original draft:** Yuxia Tang, Jian Chen.

**Writing – review & editing:** Shouju Wang.

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
