## [Decision Letter · Decision Letter 0]

8 Dec 2025

Performance Benchmarking of LLMs on China's Medical Licensing Examinations: Cross-lingual and Question-Type Effects

Dear Dr. Tang,

Thank you for submitting your manuscript to PLOS ONE. After careful consideration, we feel that it has merit but does not fully meet PLOS ONE’s publication criteria as it currently stands. Therefore, we invite you to submit a revised version of the manuscript that addresses the points raised during the review process.

Dear Authors,

Please carefully read all the comments provided by the reviewers and address them accordingly, making the necessary changes in the revised manuscript.

Best regards and keep well

We look forward to receiving your revised manuscript.

Kind regards,

Mohmed Isaqali Karobari, BDS, MScD.Endo, Ph.D. Endo, FDS, FPFA, FICD, MFDS

Academic Editor

PLOS ONE

Journal Requirements:

[This work was supported by Chongqing Natural Science Foundation (CSTB2024NSCQ-KJFZMSX0065) and the National Natural Science Foundation of China (No. 82372019), Jiangsu Province Hospital (TS202401, CZ0121002010039).].

3. Thank you for stating the following in your manuscript:

[This work was supported by Chongqing Natural Science Foundation (CSTB2024NSCQ-KJFZMSX0065) and the National Natural Science Foundation of China (No. 82372019), Jiangsu Province Hospital (TS202401, CZ0121002010039).]

[This work was supported by Chongqing Natural Science Foundation (CSTB2024NSCQ-KJFZMSX0065) and the National Natural Science Foundation of China (No. 82372019), Jiangsu Province Hospital (TS202401, CZ0121002010039).]

Additional Editor Comments:

Dear Authors,

Please carefully read all the comments provided by the reviewers and address them accordingly, making the necessary changes in the revised manuscript.

Best regards and keep well

Reviewers' comments:

Reviewer's Responses to Questions

**Comments to the Author**

1. Is the manuscript technically sound, and do the data support the conclusions?

Reviewer #1: Yes

Reviewer #2: Yes

2. Has the statistical analysis been performed appropriately and rigorously?

Reviewer #1: Yes

Reviewer #2: Yes

3. Have the authors made all data underlying the findings in their manuscript fully available?

Reviewer #1: Yes

Reviewer #2: Yes

4. Is the manuscript presented in an intelligible fashion and written in standard English?

Reviewer #1: Yes

Reviewer #2: No

Reviewer #1: This manuscript presents a timely and relevant investigation into the application of LLMs in dentistry. The framework appears robust, with notable strengths in the study design. This reviewer has only a few comments:

1. The study was conducted in one institution with one cohort, potentially introducing selection bias and limiting external validity. Please consider discussing more about how results might differ in multicenter or longitudinal designs.

2. The power analysis was good, though the number might still be a bit underpowered given the number of predictors involved. There might be a need for more justification for why this cohort represented diverse dental education contexts.

3. The formative (non-graded) nature of the assessment might not reflect high-stakes performance, where student effort could vary.

4. Prompt engineering could substantially affect AI output and consistency, so the authors’ choice of these prompts might need to be further justified, particularly with local deployments that might add another variable in the equation.

5. There seemed to be no error correction for the subgroup analyses.

6. Some of the effect sizes reported did not seem to fall under “large.”

7. Please expand the discussion further to explore the clinical/educational meaningfulness of this study.

8. The authors may consider better contextualizing their study against others that looked into the performance of LLMs in dental exams. For example, the authors might refer to:

i. Chau, R. C. W., Thu, K. M., Yu, O. Y., Hsung, R. T. C., Lo, E. C. M., & Lam, W. Y. H. (2024). Performance of generative artificial intelligence in dental licensing examinations. International dental journal, 74(3), 616-621.

ii. Chau, R. C. W., Thu, K. M., Yu, O. Y., Hsung, R. T. C., Wang, D. C. P., Man, M. W. H., ... & Lam, W. Y. H. (2025). Evaluation of Chatbot Responses to Text-Based Multiple-Choice Questions in Prosthodontic and Restorative Dentistry. Dentistry Journal.

iii. Rokhshad, R., Zhang, P., Mohammad-Rahimi, H., Pitchika, V., Entezari, N., & Schwendicke, F. (2024). Accuracy and consistency of chatbots versus clinicians for answering pediatric dentistry questions: A pilot study. Journal of dentistry, 144, 104938.

iv. Rokhshad, R., Mohammad, F. D., Nomani, M., Mohammad-Rahimi, H., & Schwendicke, F. (2025). Chatbots for conducting systematic reviews in pediatric dentistry. Journal of Dentistry, 158, 105733.

9. Please check for consistency in names and terminologies.

10. Please report all details, like SD, in the Tables.

11. Please further discuss potential AI biases.

Reviewer #2: This paper discusses an interesting topic, but there are several issues that need to be addressed. If the authors resolve these concerns, the paper has the potential for publication.

1. The abstract mentions that the accuracy of large language models in answering questions from the Chinese Clinical Practice Physician Examination “has not been systematically evaluated.” In fact, there have been many studies in this area. It is suggested to modify this phrase to highlight that the comparison of cross-lingual and question-type effects has not been sufficiently explored, which emphasizes the significance of the current study.

2. In the abstract, it is stated that Doubao-1.5-pro and ChatGPT-4 achieved the highest accuracy. However, it is unclear whether this accuracy was observed for English or Chinese question types. This should be clarified.

3. Please specify the source of the data used in the study. When referring to the “examination’s question bank,” is it an official source or from a third-party website? Also, which year's examination was used?

4. What is the total number of questions in the examination? The study used only 198 questions, how were these selected? Were they chosen randomly, or curated by clinians with some eligibility criteria? Additionally, do these questions include modalities beyond text, such as images, tables, or chemical formulas?

5. Please provide the specific dates when these models were used in the study, as well as their official release dates. This information will be useful for future comparative studies.

6. The study mentions that the authors used web UI interfaces for the models. Could you provide more details on the model parameters, such as temperature, top-p, max tokens, etc.? These are typically default settings for large language models, and providing them would be helpful.

7. It would be beneficial to include more references that are closely related to the study, such as those with PMIDs 39729356 and 38355517.

8. Is the passing standard for this exam 60% accuracy? If so, the authors should clarify in the discussion that the results of this study do not indicate that these large language models have passed the examination, since only 198 questions were used rather than the full set.

9. The models provided explanatory rationales for their answers, but the authors only calculated the accuracy of the responses and did not perform a deeper analysis. For example, completeness, rationality, hallucinations, and reasoning behind the responses from the models were not discussed. This should be addressed in the discussion section.

10. There are certain phrases, symbols, or text in the manuscript that may appear to be generated by generative AI. Could you clarify whether generative AI was used solely for text refinement, or if it was involved in other parts of the manuscript?

.

Reviewer #1: No

Reviewer #2: **Yes:** Hui ZongHui ZongHui ZongHui Zong

---

## [Author Response · Author response to Decision Letter 1]

30 Dec 2025

Prof. Yuxia Tang

Department of Radiology

The First Affiliated Hospital with Nanjing Medical University

Nanjing 210029, P.R. China

Dec 30, 2025

Dear Editor,

PLOS ONE

Thank you for your comments on our manuscript titled “Performance Benchmarking of LLMs on Chinese National Medical Licensing Educations: Cross-lingual and Question-Type Effects” (Manuscript Number: PONE-D-25-45754). We appreciate the editors and reviewers for devoting their time and effort to providing feedback and insightful comments. In accordance with the relevant suggestions, we have carefully responded to all the questions raised in the reviewers’ comments. Detailed corrections and responses are listed point by point below, and the corresponding revisions have been marked in red in the manuscript. We hope the revised manuscript meets the eligibility requirements for publication in the PLOS ONE.

Reviewer #2:

Reviewer #2: This paper discusses an interesting topic, but there are several issues that need to be addressed. If the authors resolve these concerns, the paper has the potential for publication.

1. The abstract mentions that the accuracy of large language models in answering questions from the Chinese Clinical Practice Physician Examination “has not been systematically evaluated.” In fact, there have been many studies in this area. It is suggested to modify this phrase to highlight that the comparison of cross-lingual and question-type effects has not been sufficiently explored, which emphasizes the significance of the current study.

Our response:

Thank you for this important observation. Accordingly, we have revised the statement in the Abstract as “The cross-lingual and question-type variations affecting large language models (LLMs) accuracy on the China's medical licensing educations remain insufficiently explored”. (Page 2, Lines 2-4)

2. In the abstract, it is stated that Doubao-1.5-pro and ChatGPT-4 achieved the highest accuracy. However, it is unclear whether this accuracy was observed for English or Chinese question types. This should be clarified.

Our response:

We thank the reviewer for this suggestion. We have revised the abstract as “Across all question types and languages, Doubao-1.5-pro achieved the highest accuracy at 92.0 % ± 1.3 %, whereas ChatGPT-4o had the lowest accuracy at 82.8 % ± 3.7 %”. (Page 2, Lines 11-12)

3. Please specify the source of the data used in the study. When referring to the “examination’s question bank,” is it an official source or from a third-party website? Also, which year's examination was used?

Our response:

We thank the reviewer for this suggestion. We have specified the data source in the Methods section as "The questions were curated from the official exam question pools spanning the years 2020 to 2022." (Page 3, Lines 26-27)

4. What is the total number of questions in the examination? The study used only 198 questions, how were these selected? Were they chosen randomly, or curated by clinians with some eligibility criteria? Additionally, do these questions include modalities beyond text, such as images, tables, or chemical formulas?

Our response:

We thank the reviewer for this important methodological point. The Chinese Medical Licensing Examination typically consists of 600 questions divided into three types (A, B, and C). From the official examination question bank, we employed a stratified random sampling approach to select questions proportionally from each question type. We randomly selected 198 paired English-Chinese questions with proportional representation from each question type category: 76 Type A items, 62 Type B items, and 60 Type C items. All questions were text-based single-choice questions without images, tables, or chemical formulas. (Page 4, Lines 5-12)

5. Please provide the specific dates when these models were used in the study, as well as their official release dates. This information will be useful for future comparative studies.

Our response:

We have included the study period and model release dates. We have revised the manuscript as “The study was conducted between May 13 to 20, 2025. The LLMs include ChatGPT-o3 (released April 17, 2025), ChatGPT-4o (released May 13, 2025), Gemini-2.5-pro (released March 25, 2025), Doubao-1.5-pro (released January 22, 2025), Deepseek R1 (released January 20, 2025), and Deepseek V3 (released December 26, 2024).” (Page 4, Lines 1-3)

6. The study mentions that the authors used web UI interfaces for the models. Could you provide more details on the model parameters, such as temperature, top-p, max tokens, etc.? These are typically default settings for large language models, and providing them would be helpful.

Our response:

We thank the reviewer for this valuable suggestion. In response, we have revised the methods as “All models were accessed through their official web interfaces using default parameters. Below are the input methods for different question types: for type A questions, each question was entered in a fresh conversation session; for type B questions, questions sharing the same stem were entered together in a single session, followed by a new conversation session for the next set of questions with a different shared stem. For type C questions, questions sharing the same set of options were entered together in one session, with a new conversation session initiated for the next group of questions with different shared options.” (Page 4, Lines 15-21)

7. It would be beneficial to include more references that are closely related to the study, such as those with PMIDs 39729356 and 38355517.

Our response:

We thank the reviewer for this constructive suggestion. We have added the requested references. Added References:10. Zong H, Wu R, Cha J, Wang J, Wu E, Li J, Zhou Y, Zhang C, Feng W, Shen B. Large Language Models in Worldwide Medical Exams: Platform Development and Comprehensive Analysis. J Med Internet Res. 2024 Dec 27;26: e66114. doi: 10.2196/66114. PMID: 39729356. 11. Zong H, Li J, Wu E, Wu R, Lu J, Shen B. Performance of ChatGPT on Chinese national medical licensing examinations: a five-year examination evaluation study for physicians, pharmacists and nurses. BMC Med Educ. 2024 Feb 14;24(1):143. doi: 10.1186/s12909-024-05125-7. PMID: 38355517. (Page 9, Lines 15-20)

8. Is the passing standard for this exam 60% accuracy? If so, the authors should clarify in the discussion that the results of this study do not indicate that these large language models have passed the examination, since only 198 questions were used rather than the full set.

Our response:

We thank the reviewer for this constructive suggestion. In the revised Discussion section, we have expanded the Limitations subsection as “Although the passing standard for this examination is 60% accuracy, the results of this study do not indicate that these LLMs can pass the real examination. This is because the analysis was based on only 198 questions rather than the complete examination set, and it was limited to text-based single-choice questions, excluding items containing images, tables, chemical formulas, and other non-text modalities that are typically included in the full examination”. (Page 7, Lines 14-19)

9. The models provided explanatory rationales for their answers, but the authors only calculated the accuracy of the responses and did not perform a deeper analysis. For example, completeness, rationality, hallucinations, and reasoning behind the responses from the models were not discussed. This should be addressed in the discussion section.

Our response:

We thank the reviewer for this constructive suggestion. As the primary objective of this study was to systematically evaluate the cross-lingual and question-type effects on large language models' accuracy in answering questions from Chinese medical licensing examination, our analysis focused on quantifying performance differences across these dimensions. While we acknowledge that deeper analysis of the explanatory rationales—including completeness, rationality, hallucinations, and reasoning processes—would provide valuable insights, this was intentionally scoped as a future research direction rather than the primary focus of the current study. The complexity of medical reasoning explanations requires specialized evaluation frameworks that go beyond simple accuracy metrics. To address this concern, we have revised the manuscript and added a dedicated section in the Limitations as “This study only focused on accuracy, neglecting consistency, rationale, educational value, and clinical validity. Future studies are needed to address these issues.” (Page 7, Lines 19-21)

10. There are certain phrases, symbols, or text in the manuscript that may appear to be generated by generative AI. Could you clarify whether generative AI was used solely for text refinement, or if it was involved in other parts of the manuscript?

Our response:

We thank the reviewer for the careful review and valuable feedback. We appreciate your attention to detail regarding the potential use of generative AI in our manuscript. After careful review and revision of our manuscript, we confirm that no generative AI tools were used in the writing or content development of this study. All aspects of the manuscript—including conceptualization, methodology design, data analysis, interpretation of results, and writing of all sections—were conducted entirely by the human authors.

---

## [Decision Letter · Decision Letter 1]

14 Jan 2026

Dear Dr. Tang,

We look forward to receiving your revised manuscript.

Kind regards,

Mohmed Isaqali Karobari, BDS, MScD.Endo, Ph.D. Endo, FDS, FPFA, FICD, MFDS

Academic Editor

PLOS One

Journal Requirements:

Additional Editor Comments:

Dear Authors,

Please carefully read all the comments provided by the reviewers and address them accordingly, making the necessary changes in the revised manuscript.

Best regards and keep well

Reviewers' comments:

Reviewer's Responses to Questions

**Comments to the Author**

Reviewer #1: (No Response)

Reviewer #2: All comments have been addressed

2. Is the manuscript technically sound, and do the data support the conclusions?

Reviewer #1: Yes

Reviewer #2: Yes

3. Has the statistical analysis been performed appropriately and rigorously?

Reviewer #1: Yes

Reviewer #2: N/A

4. Have the authors made all data underlying the findings in their manuscript fully available?

Reviewer #1: Yes

Reviewer #2: Yes

5. Is the manuscript presented in an intelligible fashion and written in standard English?

Reviewer #1: Yes

Reviewer #2: Yes

Reviewer #1: 1. More detailed information about the Chinese National Physician Qualification Examination, including the total number of questions (between 2020 and 2022 examinations) and the meaning of different types in its questionbank (Type A is fundational recall? Type B is Clinical reasoning?), and any details in the subjects such as oral surgery, general medicine, etc. How can the authors obtain these questions and answers? Any reason of obtaining questions between 2020 and 2022 but not more recent/older questions? Why the authors randomly select 198-paired questions? Are these English-Chinese translation official?

2. This is not a prospective study but rather a cross-sectional study.

3. Any questions that use figures/graphs/tables?

4. The result should be compared with previous studies investigating license examinations in the same and different medical license examinations

doi: 10.2196/52784

doi: 10.2196/77978

doi: 10.2196/58897

doi: 10.1016/j.identj.2023.12.007

Reviewer #2: (No Response)

.

Reviewer #1: No

Reviewer #2: No

---

## [Author Response · Author response to Decision Letter 2]

5 Mar 2026

Dear Editor,

PLOS ONE

Thank you for your comments on our manuscript titled “Performance Benchmarking of LLMs on China's Medical Licensing Educations: Cross-lingual and Question-Type Effects” (Manuscript Number: PONE-D-25-45754R1). We appreciate the editors and reviewers for devoting their time and effort to providing feedback and insightful comments. In accordance with the relevant suggestions, we have carefully responded to all the questions raised in the reviewers’ comments. Detailed corrections and responses are listed point by point below, and the corresponding revisions have been marked in red in the manuscript. We hope the revised manuscript meets the eligibility requirements for publication in the PLOS ONE.

Reviewer #1:

1. More detailed information about the Chinese National Physician Qualification Examination, including total number of questions, meaning of question types, subject distribution, how questions were obtained, why 2020–2022 were selected, why 198 pairs were sampled, and whether English translations were official.

Our response: We thank the reviewer for this constructive suggestion. We agree that a more detailed description of the examination structure and methodology is essential for understanding the study’s validity. Accordingly, we have incorporated comprehensive information in the revised Methods section (Page 3, Lines 26 to Page 4, Lines 20). The Chinese medical licensing examination utilizes three standardized formats: Type A (independent items), Type B (shared-stem items), and Type C (shared-option items). While Type A items focus on foundational knowledge retrieval, Type B and Type C formats introduce significant contextual complexity. Type B items require the model to sustain clinical reasoning across a multi-stage vignette, while Type C items assess its precision in differential diagnosis when faced with a static set of competing distractors.

Since official examination papers are not publicly released, all items for this study were sourced from widely recognized, publicly available preparatory materials provided by Beijing Medical Examination Assistance Technology Co., Ltd. We selected materials from 2020-2022 to ensure the inclusion of the most comprehensive and contemporary collections with verified answer keys. To ensure the statistical validity of this sample size, we calculated that 198 items drawn from an annual population of approximately 600 questions provide a margin of error of approximately ±5.8% at a 95% confidence interval (assuming a 50% response accuracy for maximum variance). This degree of precision is widely considered sufficient for the evaluative analysis of LLMs. To maintain consistency for cross-lingual evaluation, only text-based items were included, while questions containing images, tables, or chemical formulas were excluded. From this pool of available text-based items, we employed a stratified random sampling approach to select 198 questions, strictly preserving the original distribution of Type A, B, and C formats. This resulted in a dataset comprising 76 Type A, 62 Type B, and 60 Type C items.

The English versions were generated through a standardized forward-translation process conducted by two authors (YX Tang and SJ Wang), both of whom possess over ten years of clinical experience and have completed more than one year of formal medical training in the United States.

2. This is not a prospective study but a cross-sectional study.

Our response: We have replaced “prospective study” with “cross-sectional study” in both the Abstract sections. (Page 2, Line 5)

3. Any questions that use figures/graphs/tables?

Our response: We clarified that all included questions were text-only single-choice items, and that questions containing images, tables, or chemical formulas were excluded. This clarification has been added to the Methods section. (Page 4, Lines 12-14)

4. The result should be compared with previous studies investigating license examinations in the same and different medical license examinations

doi: 10.2196/52784

doi: 10.2196/77978

doi: 10.2196/58897

doi: 10.1016/j.identj.2023.12.007

Our response: We appreciate these helpful references. The four prior studies evaluated either a single model or only a few models, focused on a single language, or did not distinguish question types, whereas our study simultaneously assesses six LLMs across two languages and three official question types. We have incorporated all four studies into the Discussion, adding a new paragraph comparing our findings with prior evaluations of LLM performance on medical licensing examinations in China and internationally as “Our findings align with and extend previous research evaluating LLM performance on medical licensing examinations in China and internationally. Prior studies have shown that LLMs generally perform well on foundational recall questions but exhibit greater variability on complex reasoning tasks, consistent with our observation of significant question type effects [12, 14]. Other evaluations of Chinese medical licensing examinations similarly reported that English prompts may improve performance on basic knowledge tasks but offer limited benefit for higher order reasoning [15]. Additionally, recent work in dental and specialty examinations demonstrated that cross-lingual performance varies substantially across models, highlighting the importance of bilingual semantic alignment [16]. Our study contributes to this literature by providing a direct cross lingual comparison across three question types and identifying models with particularly strong cross lingual stability.” (Page 5, Lines 19)

Prof. Yuxia Tang

Department of Radiology

The First Affiliated Hospital with Nanjing Medical University

Nanjing 210029, P.R. China

Feb 13, 2026

---

## [Decision Letter · Decision Letter 2]

19 Mar 2026

Performance Benchmarking of LLMs on Chinese National Medical Licensing Educations: Cross-lingual and Question-Type Effects

PONE-D-25-45754R2

Dear Dr. Tang,

We’re pleased to inform you that your manuscript has been judged scientifically suitable for publication and will be formally accepted for publication once it meets all outstanding technical requirements.

Kind regards,

Mohmed Isaqali Karobari, BDS, MScD.Endo, Ph.D. Endo, FDS, FPFA, FICD, MFDS

Academic Editor

PLOS One

Additional Editor Comments (optional):

Dear Authors,

The authors have addressed all the reviewers' comments and suggestions, and the manuscript has undergone significant improvement. I would like to congratulate the authors and wish them all the very best in their future endeavours.

Best regards and keep well

Reviewers' comments:

Reviewer's Responses to Questions

**Comments to the Author**

Reviewer #1: All comments have been addressed

Reviewer #2: All comments have been addressed

2. Is the manuscript technically sound, and do the data support the conclusions?

Reviewer #1: Yes

Reviewer #2: Yes

3. Has the statistical analysis been performed appropriately and rigorously?

Reviewer #1: Yes

Reviewer #2: Yes

4. Have the authors made all data underlying the findings in their manuscript fully available?

Reviewer #1: Yes

Reviewer #2: Yes

5. Is the manuscript presented in an intelligible fashion and written in standard English?

Reviewer #1: Yes

Reviewer #2: Yes

Reviewer #1: Thank you. This reviewer has no further comments for this manuscript, and wishes the authors good luck with their endeavours.

Reviewer #2: (No Response)

.

Reviewer #1: No

Reviewer #2: No

---

## [Editor Report · Acceptance letter]

PONE-D-25-45754R2

PLOS One

Dear Dr. Tang,

I'm pleased to inform you that your manuscript has been deemed suitable for publication in PLOS One. Congratulations! Your manuscript is now being handed over to our production team.

Kind regards,

on behalf of

Prof Dr. Mohmed Isaqali Karobari

Academic Editor

PLOS One